# Student Perceptions of Substance Use Disorder Stigma as a Factor for Health Disparities: A Mixed-Methods Study

**DOI:** 10.3390/pharmacy11040112

**Published:** 2023-07-01

**Authors:** Rachel E. Barenie, Alina Cernasev, Kenneth C. Hohmeier, R. Eric Heidel, Phillip Knight, Shandra Forrest-Bank

**Affiliations:** 1College of Pharmacy, University of Tennessee Health Science Center, Memphis, TN 38163, USA; rbarenie@uthsc.edu (R.E.B.); khohmeie@uthsc.edu (K.C.H.); hvr987@uthsc.edu (P.K.); 2Department of Surgery, Office of Medical Education, Research, and Development, University of Tennessee Graduate School of Medicine, Knoxville, TN 37920, USA; rheidel@utmck.edu; 3College of Social Work, University of Tennessee, Knoxville, TN 37996, USA; sforres6@utk.edu

**Keywords:** stigma, substance use disorder, health profession education

## Abstract

Background: The prevalence of substance use disorders (SUDs) is an alarming problem in the United States; however, only a fraction of patients receive treatment. Stigma from both healthcare professionals and society at large negatively impacts SUD treatment. There are limited data regarding the perceptions of healthcare students on SUD stigma as a health disparity. Methods: We conducted a concurrent mixed-methods study among students enrolled in six health-related colleges at one mid-south health science center in the US over 3 months. Both an electronic survey consisting of 17 close-ended questions and researcher-led focus groups were conducted to understand their perceptions of stigma and SUDs. The research team followed the six steps recommended by Braun and Clarke regarding the data that aimed to capture associations between categories and extract and conceptualize the themes, and thematic analysis was done using Dedoose^®^ (Manhattan Beach, CA, USA) qualitative software, which facilitated all the codes being kept organized and compared the frequency of codes across categories. Results: A total of *n* = 428 students participated in the survey (response rate = 13%), and *n* = 31 students took part in five focus groups. Most student respondents, on average, either agreed or strongly agreed that: stigma currently exists in the healthcare field; stigma can lead to patients’ not receiving the appropriate care for an SUD; and stigma can lead to lower quality care provided to patients with SUDs. Two themes were identified based on the thematic analysis: (1) additional training is necessary to better equip students for addressing SUDs in practice and (2) suggestions were formed to develop synergy between didactic and clinical rotations to improve SUD training. Conclusions: It is evident that students perceive the stigma surrounding SUDs as a detriment to patient care. Opportunities may exist in professional training programs to more seamlessly and intentionally weave SUD treatment and management concepts throughout the curriculum, as well as to empower students to operate in the complex regulatory scheme that exists for SUDs in the US.

## 1. Introduction

The coronavirus pandemic has both highlighted and exacerbated health disparities in healthcare. The Centers for Disease Control and Prevention (CDC) defines health disparities as “preventable differences in the burden of disease, injury, violence, or opportunities to achieve optimal health that are experienced by socially disadvantaged populations” [1]. These disparities are also more prevalent among patients who experience certain types of conditions, such as a substance use disorder (SUD). For example, healthcare professionals are less likely to engage and empathize with patients, because of either stigma or lack of appropriate training, which reduces the quality of care these patients receive [2]. Patients with post-traumatic stress disorder who also either have food insecurity or did not complete high school are less likely to be linked to SUD care [3].

An estimated 14.5% of people aged 12 years and older have an SUD in the United States, but only a fraction receive the treatment that they need [4]. This is in part due to the stigma associated with this disease, which affects both the care patients seek and the care they receive [5,6]. Stigma is “the negative social attitude attached to a characteristic of an individual that may be regarded as a mental, physical, or social deficiency. A stigma implies social disapproval and can lead unfairly to discrimination against and exclusion of the individual” [7]. Research shows that stigma has negative impacts on SUDs, including its contribution to problematic substance use and impacts on accessing treatment and social-emotional outcomes.

For example, the treatment of opioid use disorder, one type of SUD, is markedly different among Black and white patients. Black SUD patients are more commonly managed on methadone that they must obtain from an opioid treatment program that requires daily dosing regimens, compared to white patients who more frequently receive buprenorphine treatment in an office-based setting, which is much more convenient and less stigmatizing [8]. Black patients are also more likely to claim fear of legal repercussions and mistrust of healthcare professionals and law enforcement as barriers to seeking treatment. Another example relates to the stigma associated with sexual identity and has been shown to increase alcohol craving among heavy-drinking sexual minorities [9]. Among women, Latina women report limited social support and a cultural stigma pertaining to seeking SUD treatment and are more likely to doubt the effectiveness of treatment [10].

In addition, despite copious research demonstrating the effectiveness of methadone and buprenorphine for reducing drug use and health risk behaviors and improving quality of life, there are organizations that do not support the use of these medications [11]. There are also providers who believe that the use of methadone or buprenorphine to treat OUD is merely a substitution of one drug for another rather than treatment [12]. Stigma interferes with the delivery of evidence-based treatment, and it is important that providers are educated about the effects stigma has on care for patients with SUDs.

While much research exists linking stigma to lower-quality care for patients with SUDs, perceptions of students in professional healthcare training programs about SUD stigma as a factor for health disparities is not well researched. Students in professional healthcare programs are the providers for generations to come. Thus, we aimed to characterize their perceptions, especially among racial and ethnic minoritized populations, across all health-related colleges, at one mid-south health science center in the US.

## 2. Methods

We conducted a concurrent mixed-methods study among students enrolled in six health-related colleges at one mid-south health science center in the US, including medicine, pharmacy, dentistry, nursing, health professions, and graduate health sciences [13,14]. Data collection of the quantitative and qualitative parts occurred at the same time over three months in 2021. The University of Tennessee Institutional Review Board approved this study (IRB: 21-07977-XM; 1 March 2021).

### 2.1. Quantitative Part: Survey

The survey consisted of 17 close-ended questions (Appendix B) and the topic included stigma perceptions of SUD. The survey consisted of Likert Scale questions in which the participants were asked to indicate their opinion (1 = strongly agree to 5 = strongly disagree) [15]. Some questions in the survey instrument were adapted from a previously published survey [16]. Students were recruited from all six health profession colleges at one mid-south health science center and incentivized to participate in the study exchange for the chance to win an Amazon gift card ranging from $50 to $250. The survey was delivered electronically, and all responses were anonymous. The survey responses were also captured and stored electronically. More information about the survey development and recruitment can be found in the previously published manuscript [14].

Descriptive and frequency statistics were the primary analyses used for answering the research questions in this cross-sectional survey study. Between-subjects statistics were performed based on the meeting of statistical assumptions. When normality and homogeneity of variance were met, parametric tests including independent sample t-tests and one-way ANOVAs were performed [17]. Descriptive statistics were reported for independent groups and interpreted accordingly. Post hoc testing was performed when a significant main effect was detected using Tukey’s test. Non-parametric Mann–Whitney U and Kruskal–Wallis tests were employed for group comparisons when statistical assumptions were violated. Dunn’s test was used to test for post hoc differences. All statistical analyses were performed using SPSS Version 28 (IBM Corp., Armonk, NY, USA).

### 2.2. Qualitative Part: Focus Groups

Focus groups (FGs) were selected as the primary qualitative data collection method to complement the survey and facilitate the merging and interpreting of the results. Recruitment of the participants co-occurred with the administration of the survey [15]. The FGs were conducted by two researchers (AC and SFB), and data were collected until thematic saturation was obtained. For example, the research team (AC and SFB) met after each FG and discussed the consistency of the data to ensure saturation was achieved [18]. More information about the data collection and analysis can be found in the previously published manuscript [13]. The FGs lasted, on average, 94 min. The semi-structured FG facilitators’ guide focused on different areas of the stigma associated with SUD, including: (1) Describe your experience and comfort level with these types of treatment; (2) What makes it easier to engage with a person with an SUD? What are potential barriers to engaging with a person with an SUD? and (3) What things do you think, or have you learned, that make you most successful in interacting with persons diagnosed with SUD? The emphasis of this qualitative part of the mixed methods study was to describe their comfort level with managing available treatment options for SUDs and their comfort level, including barriers and facilitators, with interactions with patients with SUDs.

All the virtual FGs were audio-recorded, and the corpus of data was transcribed by a professional service company. Thematic analysis of the data used the Braun and Clarke framework approach [19]. The research team followed the six steps recommended by Braun and Clarke regarding the data that aimed to capture associations between categories and extract and conceptualize the themes [19]. Two researchers independently read each transcript and coded, inductively, all the transcripts. The codes were clustered based on their similarities into categories [19]. The research team met multiple times to discuss the identified themes and to ensure the codes and categories developed would capture all the data. The research team discussed the similarities and differences for each emergent theme. The process of thematic analysis was done using Dedoose^®^ (Manhattan Beach, CA, USA) qualitative software, which facilitated all the codes being kept organized and compared the frequency of codes across categories.

## 3. Results

### 3.1. Survey

A total of *n* = 428 students participated in this study (response rate = 13%). Participants were 25 years old on average (Standard Deviation (SD): 4); most participants were female (*n* = 292; 79%), graduates from a 4-year college degree program (*n* = 299; 70%), not of Hispanic, Latinx, or Spanish origin (*n* = 357; 97%), and white (*n* = 248; 68%), and the primary language spoken in their home was English (*n* = 344, 94%) (Table 1). Most participants were from a college of pharmacy (*n* = 182; 49%), followed by medicine (*n* = 70; 19%), nursing (*n* = 51; 14%), health professions (*n* = 42; 11%), graduate health sciences (*n* = 21; 6%), and dentistry (*n* = 3; 1%). See Appendix A for a precise breakdown of all the demographic characteristics of the sample.

Most student respondents, on average, strongly agreed or agreed that: stigma currently exists in the healthcare field (*Mean (M)* = 1.63; *Standard Deviation (SD)* = 0.73); stigma can lead to patients’ not receiving the appropriate care for an SUD (*M* = 1.54; *SD* = 0.72); and stigma can lead to lower quality care provided to patients with SUDs (*M* = 1.57; *SD* = 0.78) (Table 1). Students also, on average, commonly reported that a healthcare professional’s negative comments, which may have been shared with the patient, student, staff member, colleague, or other individual, could impact various aspects of patient care (1 = not at all impactful; 2 = slightly impactful; 3 = moderately impactful; 4 = very impactful; 5 = extremely impactful), such as in regards to limited availability of care (*M* = 3.75; *SD* = 0.99), lower quality care (*M* = 4; *SD* = 0.94), poorer health outcomes (*M* = 4.05; *SD* = 0.96), and higher cost of care (*M* = 3.46; *SD* = 1.21) (Table 1). Student responses varied when answering the following statements: most healthcare professionals think less of a person who has been in treatment for substance use (*M* = 2.83; *SD* = 0.99); most healthcare professionals are willing to accept someone with an SUD as a patient (*M* = 2.55; *SD* = 0.8) (Table 1). Significant differences were identified between responses from men and women (*p* = 0.004), and among different races (*p* = 0.023) and colleges (*p* = 0.016) when asked whether stigma can lead to lower quality care provided to patients with SUDs, but not for other statements (Table 2 and Appendix A).

### 3.2. Focus Groups

A total of *n* = 31 participants took part in five focus groups. Most participants (*n* = 19) identified as white, and ten identified as African American. In addition, most participants (*n* = 17) were from the College of Pharmacy, while *n* = 14 were from the College of Medicine. Two themes were identified based on the thematic analysis: (1) additional training is necessary to better equip students addressing SUD in practice and (2) suggestions were developed to develop synergy between didactic and clinical rotations to improve SUD training.

Theme 1: Additional training is necessary to better equip students for addressing SUD in practice.

The first theme revealed the participants’ consensus on the need to enhance the number of lectures and practical opportunities to equip them better to prescribe medication for substance use disorders, interact with patients with SUD, and ultimately remove this obstacle to care. In the participants’ opinions, training is necessary to interact with patients and their colleagues who are misusing substances. In addition, most of the participants highlighted the need for a sustained effort to be made by various colleges to provide additional training to their students who are future clinical practitioners.

“*I think one big barrier is just the fact that you need additional training to prescribe it [the treatment MAT]. You hear about a lot of providers who legitimately feel like they don’t- like, that may not be my job to prescribe a treatment for a substance use disorder. And because it requires additional training and certifications, it’s not something that a lot of people, or as many as you would hope, do. Like what I’ve read is some surgeons may not think that it’s their role because they’re not the primary care physician. But it’s not just surgeons, it’s anything- a lot of people defer to, oh, it’s a primary care role to prescribe that, or that’s something that doesn’t initially happen in the emergency department. So I think the treatment just gets shifted and I think that’s just a big barrier, more physicians not actually being trained in treating patients.*” (FG5, P1, Medicine)

The following quotation depicts the need for training on the prescribing regulations for opioid use disorder. The participant emphasizes the obstacle for prescribers who wish to initiate treatment for SUD. The participant says:

“*I don’t know the legality of it, but I do know that there was some legal barrier to initiating suboxone treatment in-patient that I encountered a couple different times at one of the local hospitals. Again, I don’t know all the laws surrounding that and who can and cannot write that prescription, but I know that both the instances, the patient definitely would have benefited from suboxone initiation in-patient, both due to transportation barriers, getting to a suboxone clinic, and also just getting connected with a suboxone clinic.*” (FG2, P1, Medicine)

The following quotes present the participants’ concerns about the lack of training to face a situation when a healthcare professional might misuse pain medications. As one of the participants mentioned, the case might be delicate, and the training should be incorporated into their curriculum. The specific training has to be informative on how to face these challenging situations when healthcare colleagues might misuse substances, and on how to take the correct steps to correct the case.

“*I think that there absolutely should be training in how to handle it because I think in the lectures we had last semester, we were told that 10% of healthcare professionals misuse drugs at some point in their careers, and that’s a pretty big percentage. So I think that the chances of even one of us coming into contact with someone that is misusing substances is fairly high, so knowing how to handle that would be really helpful.*” (FG5, P3, Pharmacy)

“*I feel like we have training on how to professionally interact with our colleagues, but maybe not a ton on substance use disorder. And it’s mostly in the first two years of our curriculum. But I think there was a glaring lack of gravity of the situation. I think it’s hard to conceptualize- and I’m not even sure we heard many statistics about how many physicians develop substance use disorders at any point during their career. I think we could have done a better job of going through the steps of practicing having those conversations. I’m not sure we ever practiced having those conversations, but we may have just discussed the idea of having those conversations with our colleagues. And I’m not really even sure we heard a lot of statistics about substance use disorder in physicians.*” (FG5, P4, Medicine)

Although some lectures focus on the epidemiology of the disease, it would be beneficial to incorporate other aspects of SUD. As one participant says:

“*I think that [clinical scenarios or workshops] would be helpful. I think it would be definitely something more than hearing statistics in a class.*” (FG5, P4, Pharmacy)

Another participant suggested incorporating specific workshops in the last year of their medical curriculum, because they have by then been exposed to the practical sites and do not have to focus on didactic exams.

“*…For College of Medicine, maybe giving it [the workshop] in our fourth year when we don’t have exams and stuff just around the corner and where we’ve been exposed to more, so we kind of understand the gravity behind it a little bit better than in your first and second years where you’re in the classroom all day, you’re studying all day.*” (FG5, P5, Medicine)

Theme 2: Suggestions to develop synergy between didactic and clinical rotations to improve SUD training.

This theme presents various opinions on implementing multiple training pieces that would benefit the students. In addition, many participants talked about their experiences during the didactic and practical exposures and commented on how to improve their respective curricula. One participant acknowledges certain gaps in the didactic curriculum that could be addressed through practical rotations.

“*… I think for College of Medicine would be across third year in the clinical rotations is discussing how, you know, in each rotation how substance use disorder can impact those particular patients, and that way you’re kind of making it longitudinal through the third year. Having a lecture on that would be really helpful… And even beyond that, I feel making time for out-patient so you can see, whether it’s a methadone clinic or just how addiction medicine is handled, at least having people rotate through so they can see because a lot of I feel like where we’re still at is trying to change attitudes.*” (FG4, P4, Medicine)

The following quotation asserts the value of developing a synergism between didactic and clinical rotations that will benefit future students.

“*…we’ve learned like the practicality and just the pharmacology of addiction medicine, and then, in the transition in our clinical years, I think it would just be important to maybe do more than like half a day that was scheduled for me like in addiction clinic. And then, you know, like someone said, just having some more lectures. So I think the best way to decrease the stigmatization is just by exposure, so the most exposure as possible.*” (FG4, P5, Medicine)

Furthermore, the following quote also asserts the value of having opportunities through clinical rotations to work with patients with SUD.

“*…it would be great to have that, you know, as a rotational option in the P3 and P4 years, or at least, at minimum, emphasis of substance use disorders within rotations.*” (FG3, P1, Pharmacy)

Another participant emphasized the need for clinical rotations that would focus on SUD. She states:

“*…So, we [College of Pharmacy] actually have an elective course in our third year called Substance Abuse and then we have a Pharmacy Law class that goes over the different laws for narcotics and stuff like that. So, we get some exposure in pharmacy, of course, it’s not enough. So, more rotations are needed to have more exposure to patients and how to dispense methadone…etc.*” (FG2, P1, Pharmacy)

## 4. Discussion

A total of 428 students participated in the survey (response rate = 13%), and 31 students took part in five focus groups. Most student respondents, on average, strongly agreed or agreed that: stigma currently exists in the healthcare field; stigma can lead to patients’ not receiving the appropriate care for SUDs; and stigma can lead to lower quality care provided to patients with SUDs. Two themes were identified based on the thematic analysis: (1) additional training is necessary to better equip students for addressing SUDs in practice and (2) suggestions were formed to develop synergy between didactic and clinical rotations to improve SUD training. It is evident that students perceive stigma surrounding SUDs as having a negative impact on patient care, which necessitates professional training programs to more seamlessly and intentionally weave SUD treatment and management concepts throughout the curriculum, as well as to empower students to operate in the complex regulatory scheme that exists for SUDs in the US.

We found that the majority of students believe that a healthcare professional’s negative comments could impact various aspects of patient care, and focus group respondents highlighted the opportunity to create synergy between the didactic and experiential/clinical components of their training program. This aligns with the previous literature on the topic in that negative behaviors among healthcare professionals may adversely affect patient outcomes [20]. There may be an opportunity to use application-based learning, such as case studies, to enhance existing training efforts on SUDs. Since the care that people with SUDs receive is impacted by the attitudes and actions of healthcare professionals responsible for treating them, training for healthcare students requires assessment of both therapeutic knowledge and patient–provider interactions similar to those employed for smoking cessation [2].

Our study also found that students reported agreeing with the statement that most healthcare professionals are willing to accept someone with an SUD as a patient, but some providers may encounter challenges in providing them care for various reasons. One reason, in particular, is the complex regulatory scheme that exists for SUDs in the US, mainly for OUD. Recently, the federal government removed the requirement for a buprenorphine waiver (“x-wavier”), but more progress is needed, as some states still retain arbitrary statutory limitations to access SUD care requirements [21,22]. While changes to federal and state laws are cumbersome and simply not always feasible, educating healthcare practitioners thoroughly about the complex regulatory scheme that exists is a starting point to empower students to practice within the confines of the law. This is likely covered in most pharmacy curriculums in the pharmacy law course; however, when and to what extent it is covered in other professional programs is unclear. There may be an opportunity to provide additional, tailored legal education on this important topic to help future and current practitioners provide care without putting their licenses at risk.

Students reported that they believe stigma is present in healthcare; however, they also reported that there are opportunities to improve the type and delivery method of training on this topic. Feasible implementation strategies exist, which highlight the need to continue to emphasize the delivery of SUD education in curriculums. This also raises the issue of how to train practitioners who have already graduated and are in the workforce. To date, few certificate training programs exist for pharmacists to develop their skills to serve SUD patients [23,24,25,26]. Due to the emergence of new evidence, evolving practice guidelines, and more, it is important to support practitioners throughout the longevity of their entire careers.

### Limitations

This study has several limitations. First, the generalizability of our study may be limited due to its narrow geographical focus, and the results may only reflect the beliefs and educational culture of the health science center where the study was conducted. Second, only students from the medical school and pharmacy school participated in the focus groups, but students from all colleges participated in the survey component of the study. This could limit some of the recommendations for implementation to colleges of medicine and pharmacy. Third, the response rate for the survey could have been impacted due to its delivery during a scheduled break at the college.

## 5. Conclusions

It is evident that students perceive the stigma surrounding SUDs as a detriment to patient care. Opportunities may exist in professional training programs to more seamlessly and intentionally weave SUD treatment and management concepts throughout the curriculum, as well as to empower students to operate in the complex regulatory scheme that exists for SUDs in the US. Increasing students’ knowledge of and comfort with SUD treatment has the potential to reduce their stigma towards patients with SUDs and improve the quality of care their patients receive. Further research and actions are needed to reduce stigma among providers.

## Figures and Tables

**Table 1 pharmacy-11-00112-t001:** Student responses to selected questions.

	Mean	Confidence Interval	Median	SD	Range	IQR (25–75)
Stigma currently exists in the healthcare field.	1.63	1.54–1.72	1	0.73	1–5	1
Stigma can lead to patients’ not receiving the appropriate care for a substance use disorder.	1.54	1.45–1.62	1	0.72	1–4	1
Stigma can lead to lower quality care provided to patients with substance use disorders.	1.57	1.47–1.66	1	0.78	1–5	1
Most healthcare professionals think less of a person who has been in treatment for substance use.	2.83	2.71–2.95	3	0.99	1–5	2
Most healthcare professionals are willing to accept someone with a substance use disorder as a patient.	2.55	2.45–2.65	2	0.8	1–5	1
My experience at UTHSC will help me combat stigma that persons with substance use disorders may experience.	2.21	2.10–2.31	2	0.83	1–5	1
Limited availability of care	3.75	3.63–3.87	4	0.99	1–5	1
Lower quality of care	4	3.89–4.12	4	0.94	1–5	2
Poorer health outcomes	4.05	3.93–4.17	4	0.96	1–5	1
Higher cost of care	3.46	3.32–3.61	4	1.21	1–5	1
Other	3.21	3.06–3.36	3	1.25	1–5	1

**Table 2 pharmacy-11-00112-t002:** Associations between reported student characteristics and their perceptions.

	Statement (Strongly Agree = 1; Agree = 2; Somewhat Agree = 3; Disagree = 2; Strongly Disagree = 1)														
	Stigma Currently Exists in the Healthcare Field.			Stigma Can Lead to Patients’ Not Receiving the Appropriate Care for a Substance Use Disorder.			Stigma Can Lead to Lower Quality Care Provided to Patients with Substance Use Disorders.			Most Healthcare Professionals Think Less of a Person Who Has Been in Treatment for Substance Use.			Most Healthcare Professionals Are Willing to Accept Someone with a Substance Use Disorder as a Patient.		
Student Characteristics	Mean	SD	*p*	Mean	SD	*p*	Mean	SD	*p*	Mean	SD	*p*	Mean	SD	*p*
**Sex**															
Male	1.79	0.71	**0.005**	1.72	0.85	**0.004**	1.76	0.92	**0.004**	2.93	1	0.12	2.48	0.78	0.39
Female	1.54	0.7		1.45	0.66		1.47	0.71		2.74	0.96		2.57	0.85	
**Race**															
American Indian or Alaska Native	1.67	1.16	0.242	1.33	0.577	**0.01**	1.67	1.155	**0.023**	2	1.732	0.395	2	1	0.395
Asian Indian	1.62	0.81		1.48	0.68		1.57	0.746		2.95	1.024		2.29	0.717	
Black or African American	1.31	0.57		1.13	0.336		1.15	0.356		2.58	1.083		2.69	0.979	
Chinese	1.7	0.68		1.4	0.516		1.3	0.483		2.9	0.738		2.3	0.483	
Filipino	1.8	0.84		1.4	0.548		1.4	0.548		3	0.707		2.4	0.548	
Japanese	1.5	0.71		2	1.414		2	1.414		2	0		2.5	0.707	
Other Asian	1.8	0.45		2	0.707		2	0.707		3	0.707		2.4	0.548	
Other Pacific Islander	1	n/a		1	n/a		1	n/a		2	n/a		3	n/a	
Some other race	1.83	0.84		1.64	0.674		1.67	0.778		2.92	0.9		2.58	0.9	
Vietnamese	1.5	0.58		1.5	0.577		1.5	0.577		2	1.414		2	0	
White	1.63	0.72		1.57	0.761		1.6	0.577		2.81	0.941		2.57	0.824	
**College**															
Dentistry	1.33	0.58	**0.16**	1.33	0.58	0.1	1.33	0.58	**0.016**	3	1.73	0.09	1.67	0.58	0.066
Graduate Health Sciences	1.81	0.81		1.52	0.68		1.57	0.75		2.38	1.12		2.81	1.03	
Health Professions	1.86	0.78		1.79	0.72		1.93	0.75		2.98	0.84		2.45	0.67	
Medicine	1.58	0.7		1.56	0.77		1.51	0.78		2.83	0.87		2.39	0.73	
Nursing	1.36	0.63		1.42	0.76		1.44	0.79		2.52	1.02		2.54	0.95	
Pharmacy	1.58	0.68		1.44	0.67		1.46	0.75		2.82	0.98		2.63	0.83	

## Data Availability

The data presented in this study are available in the Appendix A.

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
