# Peer review of "Student Perceptions of Substance Use Disorder Stigma as a Factor for Health Disparities: A Mixed-Methods Study"

_pharmacy, 2023, doi:10.3390/pharmacy11040112_

Round 1

Reviewer 1 Report

Introduction: Great information here, however the flow is disjointed and seems a bit scattered and needs to be reorganized/re-written.  

Paragraph starting line 48, talks about the stigma of disease  but then the example is illustrating a difference based on race and not based on the disease.  If focusing on racial disparities would address that as a separate paragraph.

Paragraph starting line 59, unclear why you are just defining stigma here when it is mentioned earlier.  Also, is this something that needs to be defined?  Seems like it would be unnecessary for the audience.

Line 68 - give examples of the opioid agonists used in the study.  Was unclear if it was just methadone or if buprenorphine had the same stigma as a partial opioid agonist.

Statement line 76 - other than providers not wanting to utilize "opioid agonists" for treatment of OUD, are there other examples that could be provided?  Not many were shared in the introduction.  

Line 77 "perceptions of students in .....about stigma as a factor for health disparities" - this does not indicated SUD.  Need to clarify the objective of your study.

Methods:

Would be helpful

Sentence, Line 94: "the survey was peer-reviewed..." is this necessary?

Paragraph starting line 101: Frequency and percentages are types of descriptive statistics.  If stating that descriptive statistics were used, then you should not name these separately.

Results:

Line 142: include how many students to which the survey was sent

Line 151: "stigma currently exists in the healthcare field" - add for SUD

Line 155: Please clarify this.  If the higher the number means they "disagreed" with a statement, is seems that they did not agree that lower quality of care or poorer health outcomes is likely in these patients

Line 163: Can you clarify what was compared for the P=0.023 for "among different races"?  There are 15 different races listed so it is unclear what the trend was with this p-value and exactly what the p-value was evaluating.

Same comment above for the "colleges" p = 0.016 - there were 6 programs so this comparison should be broken down further to understand what was compared on that specific question.

Focus groups section: I think all of the individual comments should be deleted and only the coded themes presented. It seems inappropriate and may be open to bias to just present handpicked comments.

Table 2 - type in row 3, disagree = 4 and strongly disagree = 5

Column "range" - what is the number in that column?  It is a single number so would not qualify this as a range.

Table 3

Really had a hard time reading this as it is currently formatted. Seems like there are random p-values thtat are not clearly comparing anything.

Discussion

Line 293 - multiple places a "healthcare professional's negative comments" is mentioned.  But should clarify this.  Are these comments to the patients?  Or is it more their negative view?

Line 302-305 - Clarify the statement that was asked in the survey.  Did the survey question ask about "additional challenges" - the way this sentence reads, it sounds like it did.  Also - most SUD should be done by professionals with proper training when possible - are you referring to the challenges of treating the patient in general or for treatment of SUD.  I would not infer the latter in reading your original statement so I would not make that leap unless that is what you asked.

Line 316 - again - be specific that this is stigma of SUD.  

Overall the use of English language is fine but the quality of writing is poor and needs a major edit in multiple areas.

Author Response

Please see the file attached

Reviewer 2 Report

Dear Authors,

thank you for the opportunity to review this paper. Please find below my comments:

1) Despite the conducted normality and homogeneity of variance analyzes, the Likert scale is a type of ordinal scale (the data are in order but they do not have a “real” representation in the real world unlike parameters like height or age, for instance) instead of interval or ratio scale (where data have representation in real world). For data on ordinal scale, non-parametric tests should only be used.

2) You mentioned that thematic analysis was conducted according to Braun and Clarke. However, the paper contains inconsistencies with the Braun and Clarke approach – such as saturation (how can you be sure it was achieved while still collecting data?), “emergent theme” phrase – when Braun and Clarke explicitly emphasized on numerous occasions including their 2006 paper that data do not simply “emerge’ like Venus on the half shell” in the famous Botticelli painting, or comparing frequencies of codes. Apart from looking once more on Braun and Clark’s 2006 paper, please also check their later papers where they explain some of the misconceptions on their method that arose over the years:

Braun V, Clarke V. Thematic analysis. In: APA handbook of research methods in psychology, Vol 2: Research designs: Quantitative, qualitative, neuropsychological, and biological. Washington: American Psychological Association; 2012. p. 57–71.

Virginia Braun & Victoria Clarke (2019): To saturate or not to saturate? Questioning data saturation as a useful concept for thematic analysis and sample-size rationales, Qualitative Research in Sport, Exercise and Health, DOI: 10.1080/2159676X.2019.1704846

Braun V, Clarke V, Hayfield N, Terry G. Thematic Analysis. In: Handbook of Research Methods in Health Social Sciences. Singapore: Springer Singapore; 2019. p. 843–60.

Braun V, Clarke V. One size fits all? What counts as quality practice in (reflexive) thematic analysis? Qual Res Psychol. 2021;18:328–52.

Braun V, Clarke V. Toward good practice in thematic analysis: Avoiding common problems and be(com)ing a knowing researcher. Int J Transgender Heal. 2023;24:1–6.

3) What was the average duration of focus groups?

4) Please modify your Tables (especially Table 3) because it is hard to read when numbers skip to the next line. You could use the other design (full-width) in the Journal’s template.

5) I do not really get the difference between the 1 and 2 theme. The second theme also focuses on the additional training needs of students so how is it different from the first one? So I think you should explicitly explain the differences/show how they are different or the thematic analysis should be done again.

6) The Discussion section seems limited in terms of the comparisons with the available literature published so far.

Author Response

Please see the file attached

Reviewer 3 Report

The manuscript contains data that may be of interest, although due to the theme it would seem more appropriate in a psychology or addiction journal, especially if we assess the type of data studied and the methodology used. Below I detail some aspects for the authors to take into account in an in-depth review of the current version:

- The number of participants in the study is small (n= 428) and it should be justified why the response rate was only 13%. One limitation to comment on is the great bias of female participants (n= 292), an aspect that may be biasing the results. The % of women should be reviewed, which is indicated to be 79% when it should be 68%? Furthermore, Table 1 shows only 74 men, in addition to 3 who preferred not to respond, which does not add up to the total number of participants who claimed to have participated in the abstract and method. The data of the quantitative study should be presented both for the total sample and for men and women, and explore whether there are sex differences. If so, they will be conveniently discussed and if not, the data will be of great interest.

- Statistical analysis should be limited to non-parametric tests. There are no continuous variables in the strict sense (interval scales) and the approach or presentation of results is not adequate (Table 2). It should also be checked that on the Likert scale of response to the survey it is from 1 (strongly agree) to 5 (strongly disagree) but in the Table the data corresponding to 4 and 5 are indicated as 1 and 2. In general, there is considerable laziness in the presentation of numerical data both in text and in tables.

- Table 1 can be greatly improved, the categories without cases (0) should be eliminated (Chamorro, Korean, Native Hawaiian ans Samoan). And since the analyzes must be carried out again, the authors are advised to consider the elimination of participants from very minority categories such as Japanese or other Pacific islander, as well as dentistry students. This same table should detail the characteristics of the 31 participants in the focus groups.

 - An in-depth review of the text is required as it contains numerous fonts. In the section of references from 8 appears the numbering with Romans, which has also been linked in the text with this format. In addition to this review, the authors should incorporate more literature in relation to the theme of the manuscript (the current one is not in a homogeneous format according to the normal format of the Journal and it can be improved in terms of choice). There are numerous and excellent recent publications, which if considered will allow the discussion to be reworked with greater foundation and approach to theoretical and applied aspects of interest with less repetition of the results obtained.

Author Response

Please see the file attached

Reviewer 4 Report

The authors investigated health students’ perceptions on whether and patients with substance use disorders are prone to lower-quality health care. They conducted a survey, where they found that participants are aware that stigma exists may be a problem for health care. They also made focus groups, whose output related to the need to increase training and synergies between clinical practice and the education system.

The paper contains useful information, mostly to the health care system- and is reasonably well-written, but some points need to be addressed for increased scientific soundness.

Abstract

The expression health disparity is not understandable at this point.

Intro

Lns 40-42 Clarify which comparison group the word “differences” refers to. SUD patients vs?...

Why study students in professional healthcare programs? I think I can understand why, but it could be important to make it clear at this point: why is this a gap/research need?

Methods

Why did the authors not generate a global score for the questionnaire (or 2 scores, reflecting the two blocks of awareness of stigma+awareness of stigma impact). The analysis question-by-question is a bit opaque.

If the authors analyse item by item, why not compare scores across questions to give a glimpse on the most consensual/critical facets of this problem?

Results

Tables (mostly table 3) are not well formatted and it is very hard to understand the contents.

Please specify the effects of gender, race and college on the question where these effects were observed. There were many categories in race and college, so we need post-hoc comparisons.

Why are there only 11 questions out of the 17?

Discussion

The effects (or lack of) demographic effects on the questionnaire should be explored, other wise there is no point in providing sociodemographics with such detail.

Author Response

Please see the file attached

Round 2

Reviewer 1 Report

No additional comments; addition of appendices is helpful.

Reviewer 2 Report

Dear Authors, 

Thank you for implementing my suggestions. 

Reviewer 3 Report

The authors have made some changes that have improved the manuscript, but they have not addressed all the issues I raised. One of the essentials is to have kept the original analyzes when the working data do not constitute interval scales. Your response letter is very poor, responding to only one item that each point I wrote contained and without any justification.